# Evaluation of Mother’s Stress during Hospitalization Can Influence the Breastfeeding Rate. Experience in Intensive and Non Intensive Departments

**DOI:** 10.3390/ijerph17041298

**Published:** 2020-02-18

**Authors:** S. Foligno, A. Finocchi, G. Brindisi, A. Pace, P. Amadio, I. Dall’Oglio, A. Portanova, P. Rossi, A. Dotta, G. Salvatori

**Affiliations:** 1Academic Pediatrics Department, Bambino Gesù Children’s Hospital, Piazza di Sant’Onofrio, 4, 00165 Rome, Italy; andrea.finocchi@uniroma2.it (A.F.); alessia.pace1992@gmail.com (A.P.); paolo.rossi@opbg.net (P.R.); 2Pediatrics Department, Umberto I Hospital, Sapienza University, Viale Regina Elena 324, 00161 Rome, Italy; 3Medical and Surgical Intensive Care Unit, Bambino Gesù Children’s Hospital, Piazza di Sant’Onofrio, 4, 00165 Rome, Italy; patrizia.amadio@opbg.net (P.A.); anna.portanova@opbg.net (A.P.); andrea.dotta@opbg.net (A.D.); guglielmo.salvatori@opbg.net (G.S.); 4Professional Development, Continuing Education and Nursing Research Service, Bambino Gesù Children’s Hospital, Piazza di Sant’Onofrio, 4, 00165 Rome, Italy; immacolata.dalloglio@opbg.net

**Keywords:** breastfeeding, stress, hospitalization

## Abstract

A mother’s milk offers several benefits to infant’s health, but, some factors may lead to a reduction in the duration of breastfeeding, such as maternal stress. The objective of our study is to determine if the rate of breastfeeding can be influenced by stress induced by infants’ hospitalization. A preliminary observational non-randomized study was carried out in Bambino Gesù Children’s Hospital between October 2016 and January 2017, in order to elucidate a possible relationship between breastfeeding and maternal stress, linked to hospitalization. We modeled the modified version of the PSS NICU (Parental Stressor Scale—neonatal intensive care unit) questionnaire, which investigated parental stress during hospitalization. This included 33 items with a score from 0 to 5. The overall score, high stress, was established at 85 points or higher. The principal statistically significant correlation was between ‘high PSS score’ and reduced breastfeeding during hospitalization (*p*-value: 0.048; OR: 2.865, 95%; CI: 1.008–8.146). This relation was not influenced by other descriptive characteristics of the mother. The PSS questionnaire can be an instrument to evaluate the influence of stress in breastfeeding and to monitor the rate and success of lactation. Our study highlights that the stress from hospitalization could influence the success of breastfeeding, mostly in intensive settings and during long hospitalizations.

## 1. Introduction

The American Academy of Pediatrics recommends exclusive breastfeeding for about 6 months, with continuation for one year or longer as mutually desired by mother and infant [1]. A mother’s milk offers several medical and neurodevelopmental benefits for health over short and long periods [2]. The protective effect is the result of the combination of duration and exclusivity of breastfeeding. Some factors may lead to a reduction of the duration of breastfeeding whether exclusive or partial, such as the advanced age of mothers, being nulliparous, the modality of delivery, low socioeconomic status (depending on settings), numbers of peers and parental relationship [3,4,5]. Breastfeeding can be influenced by chronic or acute stress, that could have an immediate or delayed effect [6]. Infant’s hospitalization is a significant barrier against starting and continuing breastfeeding [7,8,9]. During this time, the mother may develop anxiety and fears that could influence the success of breastfeeding. Medical staff and nurses should not only focus on child care, but also support, promote and protect breastfeeding, to create a safe and comfortable environment [7,10]. In this perspective, breastfeeding and Kangaroo mother care (skin to skin contact) could help mothers to relax and subsequently, to take care of their own child [11]. This can be possible only by following a physiological breastfeeding process, which means stimulating suction practice, sustaining milk production, and therefore establishing a positive cycle [12]. Breastfeeding is an interactive process between mother and child and it is regulated by two main hormones: Prolactin (PRL) and oxytocin (OT), through an autocrine mechanism by the feedback inhibitor of lactation (FIL). Breastfeeding is a mechanism of demand and offer, where the suction activates a central reflex inducing PRL and OT production which allows milk ejection (ejection reflex). It is estimated that 85% of mothers breastfeed after birth, and over 97% continue with success when provided suitable support [13,14]. During infant’s hospitalization, parental stress rises, especially in intensive care and in the neonatal intensive care unit (NICU) [15,16,17]. The objective of our study is to determine if the rate of breastfeeding can be influenced by maternal stress induced by the infant’s hospitalization. The study was carried out in different departments to understand if the intensive setting is the main cause of mother’s stress. Additionally, we studied how maternal stress was influenced by the level of information and education provided by medical and nursing staff during the hospitalization.

## 2. Material and Methods

### 2.1. Study Design and Population

This preliminary observational epidemiological non-randomized study was performed in the Bambino Gesù Children’s Hospital, in order to elucidate a possible relationship between breastfeeding and maternal stress linked to hospitalization. To fully consider the potential stress factors associated with the hospital setting and environment, we provide an overview of how our medical departments and facilities are organized.

The Paediatric University Department of Bambino Gesù Children’s Hospital occupies three different floors. On the ground floor, there is the semi-intensive area equipped with single rooms; the first floor is for the infectious and immunological departments, also equipped with single rooms for infectious disease isolation, and; the second floor is for the pediatric general wards, equipped with twin rooms. In general, the level of complexity and severity of the assistance required decreases from ground to second floor.

Considering the neonatal intensive unit, the intensive ward in Bambino Gesù Children’s Hospital is not organized as a birth center; there is no rooming-in, so there are no single rooms for mother and child. This setting could contribute to raising parental stress.

The inclusion criteria selected for mothers of newborns or infants who were exclusive, complementary or predominantly breastfed, and admitted to Bambino Gesù Children’s Hospital in the departments of neonatal intensive care and semi-intensive children care and pediatrics general ward, between October 2016 and January 2017. The exclusion criteria disqualified mothers who never experienced breastfeeding. Reviewing the existing literature, we studied different scales that evaluated the parental stress levels. On the basis of these scales, we modelled the modified version of the PSS NICU questionnaire (Parental Stressor Scale—neonatal intensive care unit) that investigated parental stress during hospitalization. This study was performed following good clinical practice (GCP), according to the Helsinki Declaration, and in line with the Ethics Committee of our Hospital. For this type of observational study, was sufficient the notification to the Ethics Committee of Bambino Gesù Children’s Hospital. Thereafter, the study may start 60 days after notification using the silence / assent procedure.

Participants were asked to complete the questionnaire during hospitalization and within 72 h before discharge. These specific points in time were selected to record the quality of mother’s stress during the period of hospitalization. To assess parental distress, the PSS NICU was developed from the Parental Stressor Scale: Pediatric ICU (PSS: PICU) by Carter (1985) [18] and Miles (1983) [19]. It was further refined by Miles et al. in 1993 [20], to measure the parental perception of stressors due to the physical, psychosocial and emotional environment of the neonatal intensive care unit. In the current study, a modified version of the PSS NICU [21] was presented in the original questionnaire [22], reformulating the English questions in a more straightforward Italian version, suitable to all educational levels. The validity and internal consistency of the scale has been confirmed in the USA [23] and the UK [22]. The questionnaire included 33 items, and was structured in three parts: Sights and sounds subscale; look and behavior subscale, and; communications with staff. A score from 0 to 5 was given to each item, referring to the stress perceived by the mother. The overall score which we decided to define as “high stress” was established at 85 points or higher, in order to create statistical correlations. The stress scale was interpreted by generating a summary of the score from each item. Those scores recorded as 0 were either “not applicable” or “not important” to the mother.

### 2.2. Statistical Analysis

For the statistical analysis, a software statistical package for social sciences was employed (SPSS Inc., Chicago, IL, USA). The baseline characteristics of patients were presented as percentages in the case of categorical variables, and as mean ± SD of the average in the case of continuous variables. The categorical variables were compared by the Pearson χ2 test. As for non-categorical variables, the Student *t*-test was used. The statistically significant parent survey variables associated with each subscale score were then included in a stepwise multiple regression to predict each subscale score. Logistic regression was used to calculate odds ratios of parent survey variables and group attendance. Statistical significance was considered for *p*-values of equal to or less than 0.05.

## 3. Results

From October 2016 to January 2017, the study enrolled 87 mothers of infants or newborns admitted to the Bambino Gesù Children’s Hospital, who breastfed exclusively, complementarily or predominantly. While we offered the questionnaires to 100 mothers, only 87 mothers gave their consent to participate in the study. The general descriptive features of the mothers are reported in Table 1 and Table 2, while the characteristics of children are reported in Table 3, and Figure 1 and Figure 2.

The average age of mothers was 33.24 (20–44 years old) of which 45 (51.7%) did not breastfeed exclusively, while 42 (48.3%) breastfed exclusively. Moreover 48.3%, approximately half of the total population, introduced infant formula during hospitalization.

Concerning the support, promotion and protection of breastfeeding, 24% of the total population did not receive any information or education about breastfeeding at all; 62 mothers (71.3%) received limited information and education and nothing about benefits of breastfeeding; 82.2% of the total population reported an uncomfortable environment in the Hospital for practicing breastfeeding. Furthermore, 60.9% of mothers said that no help was given for issues linked to breastfeeding during hospitalization, and 52.9% reported the sensation of reduction in milk production. This result was confirmed by objective data extracted from clinical medical reports, where 66.7% of the total population during hospitalization breastfed less than when the child was admitted.

Table 4, below, shows the results regarding the relationship between breastfeeding and stress as extracted from the modified PSS.

The score “high PSS” is correlated to the reduction of breastfeeding during hospitalization. Through a single linear regression, a significant relation between the two variables above was found (*p*-value: 0.048; OR: 2.865, 95%; CI: 1.008–8.146). Applying a multiple linear regression, this correlation was not influenced by subjective variables (mother’s age, nationality, profession, education, socio-economic status, relationship, previous pregnancy, previous breastfed newborns) with a significant correlation (*p*-value: 0.002; OR: 0.207, 95%; CI: 0.53–0.8). So the variable itself (less breastfed during hospitalization) was related with “high stress score”. This relation was not influenced by information given to the mothers during hospitalization (*p*-value: 0.025; OR: 3.6 95%; CI: 1.74–11.4). No correlation was found between “moderate or high stress” and “sensation of milk reduction”. The intensive setting (semi and intensive neonatal unit) presented a higher number of stressed mothers than other departments. Overall, applying a single linear regression between “high PSS scores” and departments, a strong correlation with the admission to NICU was noted (*p*-value: 0.003; OR: 0.225, 95%; CI: 0.083–0.609). Finally it was detected a significant relation between the duration of hospitalization (above of 20 days) and “high PSS score” (*p*-value: 0.028; OR: 3.54, 95%; CI: 1.148–10.955). To check the influence of each part of PSS on breastfeeding rates, was conducted a linear regression between each part of the PSS and “less breastfed”. The third part of PSS questionnaire had a significant correlation with “less breastfed” (*p*-value: 0.039; OR: 1.042, 95%; CI: 1.0–1.08). This part of PSS questionnaire referred to parental role.

## 4. Discussion

Stress is the organism’s total response to environmental demands or pressures. There are different types of stress, negative and positive. The first one occurs when stressful stimuli lead to a progressive strain, until a break of psychological and physical barriers, while positive stress or eustress is a complex of stimuli that trains the ability to adapt [24]. Eustress represents a type of energy utilized to reach a goal and so this stress helps to make personal improvement. One of the best examples of adaptation to stress is breastfeeding. Indeed, oxytocin contributes to mother’s relaxation and promotes a sense of childcare, while skin-to-skin contact facilitates PRL release [25,26]. Beta-endorphins are hormones of pleasure reward, and childcare contributes to an increase in their levels. This also could represent a central motivation of pleasure [27]. Moreover, beta-endorphins are released from human milk and swallowed by the child, representing a stimulus of reward for the child [27]. Stress caused by an irrational fear of insufficient breastfeeding can lead to depressive disorder [28], which is one of the most common causes for not starting or continuing with breastfeeding exclusively [29,30]. The mother’s milk becomes more important in the case of a newborn, in particular those who are admitted to the hospital, especially to intensive care unit. In fact, it reduces the risk of infectious and intestinal diseases and helps to fortify the immunological system of these weak patients. Moreover, it creates a unique relationship, favored by skin-to-skin contact. Only one study in the literature has showed social barriers as a stress factor capable to reduce the commencement of breastfeeding [31]. No study has demonstrated the influence of maternal stress on lactation, caused by hospitalization. In contrast to other studies, in our report, breastfeeding was not influenced by mother’s age, smoking habit or previous pregnancy. Degree of education and profession were not linked to maternal stress, which is different to the data showed in the literature [32]. As highlighted by Holmes et al. (2013) [33], the importance of education and information about breastfeeding both before and during pregnancy is critical. In our study, 62 mothers (71.3% of total) reported that no information or education on the benefits of breastfeeding was provided to them. Nevertheless, no significant correlation was found between lack of information and “high PSS score”. Therefore, it is critical that an effective communication with mothers is conducted [34]. In our study, 60.9% of the mothers had no help to solve and cope with problems related to breastfeeding. During hospitalization, a number of studies highlighted that parental stress rises, particularly in intensive care setting. Several studies validated and reported the utilization of PSS, adapted to different settings, namely, general pediatric unit (GPU), pediatric intensive care (PIC) and neonatal intensive care unit (NICU) [15,17,20]. Our goal was also to analyze the influence of different settings on mother’s stress levels. In the study conducted by Turner et al. (2015) in a NICU, the results showed that mothers were stressed in all fields on the parental stress scale (sight and sound, baby behavior and relation and role of parent) [35]. Similar to Turner’s analysis, we found a relation between “high PSS score” and the intensive care setting. From our study, we reported that the “high PSS score” of the third part of the PSS questionnaire indicated a strong relation to a reduction in breastfeeding. Additionally, it was reported in another study that a “high PSS score” in cases of long hospitalization (more than 20 days) indicated increased stress that was linked to seven days or more of hospitalization [32]. The principal statistically significant correlation was between “high PSS score” and reduced breastfeeding during hospitalization (*p*-value: 0.048; OR: 2.865, 95%; CI: 1.008–8.146). This relation was not influenced by other descriptive characteristics of the mother. There are some limitations to this study, including that the population was not uniform (different age and basal characteristics). Nevertheless, this population still provides valuable information about the environment and the different settings. Another limitation is the language barrier, where some units of the population did not have full comprehension of the Italian language. It is important to state that this was a preliminary pilot study and additional follow-up studies may be necessary.

## 5. Conclusions

The data from our study show that in a specialized pediatric hospital, the level of information and education regarding the benefits of breastfeeding was insufficient. Furthermore, there was a lack of necessary support to overcome the difficulties associated with hospitalization, which was correlated to a reduction in breastfeeding. Improvements are needed to assure an increase in the percentage of breastfeeding mothers during hospitalization. It is an aim and priority of WHO/UNICEF (World health organization/ United Nations Children’s Fund) initiatives to promote the frequent training and distribution of information on protection, promotion and support of breastfeeding to mothers. In this scenario of hospitalization, the implementation of a shared protocol between nurse and pediatrician to protect breastfeeding is fundamental. A dedicated staff and a comfortable environment should be the standard of care for all mothers. As for our findings regarding the third part of the PSS questionnaire, it seems important to place the maternal role at the center of care during hospitalization. Our study highlights that the stress from hospitalization could influence the success of breastfeeding. This is particularly true in intensive care settings, and during long hospitalizations. Breastfeeding creates a unique relationship between mother and child, thus reducing the negative effects linked to hospital stress. Finally, this study showed that the PSS questionnaire can be an instrument to evaluate the influence of stress in breastfeeding and may be used to monitor the rate and success of lactation in intensive care settings.

## Figures and Tables

**Figure 1 ijerph-17-01298-f001:**
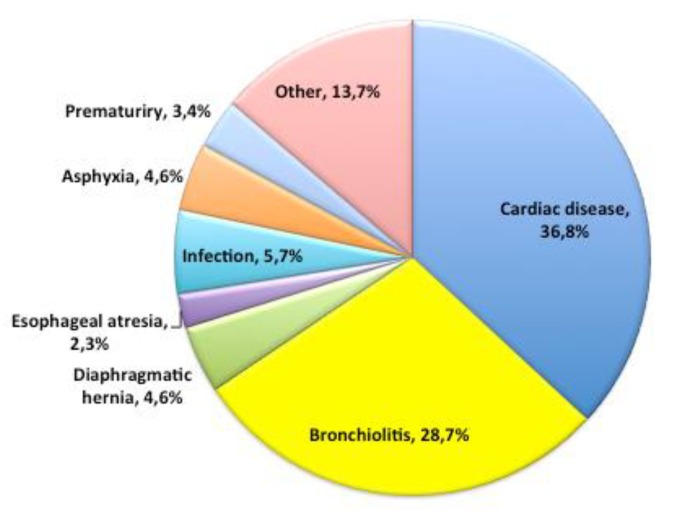
Percentage of diagnostic causes of admission of the general population.

**Figure 2 ijerph-17-01298-f002:**
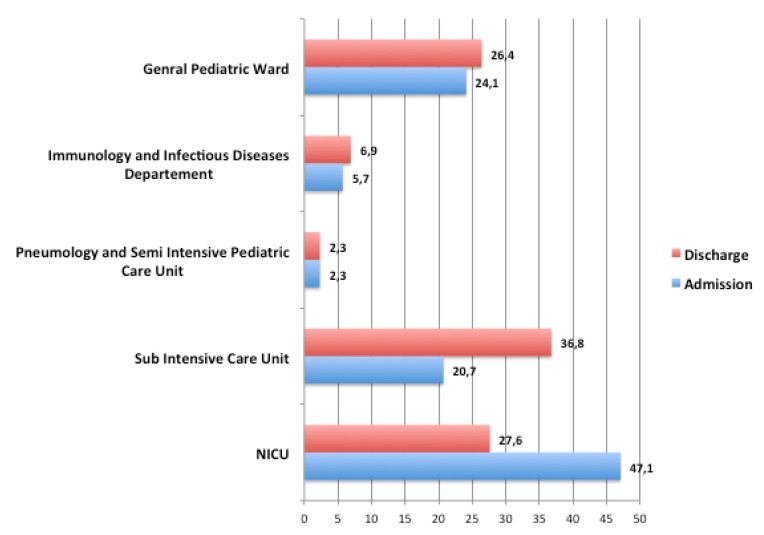
Percentage of population divided in the different departments.

**Table 1 ijerph-17-01298-t001:** Descriptive characteristic of mothers.

	Min	Max	Median	DS
**Age**	20	44	33.24	5.025
**Number of Newborns**	1	9	1.77	1.053
**Number of Breastfed Newborns**	1	4	1.66	0.729
**Previous Pregnancy**	0	9	1.02	1.455

**Table 2 ijerph-17-01298-t002:** Descriptive characteristic of mothers.

**Nationality**	
Italian	67 (77%)
Foreign	20 (23%)
**Education**	
Elementary school	1 (1.1%)
Medium school	46 (52.9%)
Degree	40 (46%)
**Occupation**	
Unemployed	34 (39.1%)
Employed	40 (46%)
Freelance	5 (5.7%)
Laborer	8 (9.2%)
**Socio-Economic Status**	
Low	9 (10.3%)
Medium	71 (81.6%)
High	7 (8%)
**Relationship between Mother and Father**	
Split up	30 (34.5%)
Married	56 (64.4%)
Live together	1 (1.1%)
**Previous Pregnancy**	
Yes	48 (55.2%)
No	39 (44.8%)
**Type of Birth**	
Natural	44 (50.6%)
Caesarean section-Epidural	38 (43.7%)
Caesarean section-General anesthesia	5 (5.7%)
Smoke during pregnancy	
Yes	12 (13.8%)
No	75 (86.2%)
**Smoke during Hospitalization**	
Yes	69 (79.3%)
No	18 (20.7%)

**Table 3 ijerph-17-01298-t003:** Descriptive characteristic of children’s population.

	Min	Max	Average	SD
**Gestational Age at Birth (g)**	23	39	31	11.31
**Weight of Birth(g)**	760	4100	2430	2361.74
**Weight at Admission(g)**	767	6000	3383.5	3700.3
**Weight at Discharge(g)**	2120	6540	4330	3125.42
**Hospitalization (days)**	7	40	23.5	23.33

**Table 4 ijerph-17-01298-t004:** The table shows the maternal stress as measured by PSS (*n* = 87).

Subscale and Components	Median	SD
The presence of monitors and equipment	2.20	1.421
The constant noise of equipment	2.78	1.434
The sudden noise of monitor alarms	3.32	1.521
The other sick babies in the room	1.85	1.394
The large number of people working in the unit	1.30	1.036
Having a ventilator breath for my baby	2.33	1.853
Bruises and cuts on my baby	1.87	2.067
Unusual color of my baby	1.49	2.011
Unusual breathing of my baby	2.30	2.075
Seeing my baby without breath	2.01	2.127
The small size of my baby	1.99	1.908
Tubes and equipment on or near my baby	2.62	1.812
Seeing needles and tubes put in my baby	2.07	2.128
When baby seemed to be in pain	3.70	1.699
When baby looked sad	3.67	1.420
The limp and weak appearance of my baby	3.46	1.724
Being separated from my baby	3.09	2.072
Not feeding my baby myself	2.77	2.106
Not being able to care for my baby myself	2.29	2.023
Not being able to hold my baby when I want	2.59	2.165
Sometimes forgetting what my baby looks like	0.86	1.571
Not being able to share my baby with other family members	2.26	1.728
Feeling helpless and unable to protect my baby from pain and painful procedures	3.37	1.818
Being afraid of touching or holding my baby	1.91	1.762
Feeling staff are closer to my baby than I am	1.68	1.762
Feeling not useful to help my baby	2.82	1.775
Feeling staff do not explain in a clear way	2.09	1.783
Feeling staff explain a different clinical situation to reality	2.13	1.605
Having relation with a several different professional figures (nurse, doctors, interns)	1.90	1.414
Feeling staff does not give information about the clinical status of my baby	2.78	2.088
Feeling staff do not know the real needs of my baby	2.52	1.934
Feeling sad about leaving my baby alone	2.69	1.937
Feeling sad because my baby is in hospital	4.69	0.556

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
