# Peer review of "Evaluation of Mother’s Stress during Hospitalization Can Influence the Breastfeeding Rate. Experience in Intensive and Non Intensive Departments"

_ijerph, 2020, doi:10.3390/ijerph17041298_

Round 1

Reviewer 1 Report

The manuscript is very original and gives important news about the relationships between the breastfeeding and the stress. The possibility to use a specific questionnaire to evidence stress and related interferences on the breastfeeding offers a good opportunity to help the parents in this critical period of the family life. The only criticism that I can do on the maniuscript, is related to the number of subjects participating in the study. A major number of subjects could offer a better statistical evidence of the results. I appreciate the conclusion of the manuscript in which thre is the invitation to give more care in the psychological approach to the parents.

Author Response

Thanks a lot for your suggestions and your notes. Absolutely a larger sample would have provided greater significance to our results. It will be done in future.

Reviewer 2 Report

Please review the entire paper for punctuation errors and misspelling of words

The graphs and not in the correct format

Please see link below for correct formatting of tables and graphs

Review the references to ensure they are in the correct APA format

https://owl.purdue.edu/owl/research_and_citation/apa_style/apa_formatting_and_styleguide/general_format.html

Line 21 comma is needed after but

Line 24 missing the word after infant's

hospitalization is spelled wrong Line 24, 61, and results section

Modelled is spelled wrong

Line 33 remove the word "the after stress

Line 79 center instead of centre

Line 80- should be raising

Line 96 comma before and

Line 105 should it be items or item

Lines 71 and 83 Semi-Intensive need a dash

Line 43 socio-economic should be one word

Line 120 should be questionnaires

Line 105-should be items or item

Line 96 add a comma before and

Line 81should it be exclusively or exclusive

Line 111 is this in the correct  (PersonX2 test)

What is PSS acronym

The results section need corrections please see updates needed

Author Response

Line 21: comma has been added as suggested 

Line 24: "infants' hospitalization"

Line 26: corrected

Line 33: "The" removed

Line 61: corrected

Line 79: done

Line 80: corrected

Line 96: done

Line 105: done

Line 71-83: done

Line 43: done

Line 120: done

Line 81: exclusive

Line 111: corrected

PSS: Parental Stressor Scale

I have done the review for misspelling words, punctuation mistakes, references (APA format), tables and graphs.

I have corrected also the result section. 

Thanks a lot for your suggestions.

Reviewer 3 Report

I suggest a thorough review of the language style

I also suggest review of references using the same style for all

Specific suggestions are in the attached document

Author Response

I have done the review of the language style, references and all the other modifications suggested in the PDF file.

Thanks a lot